# Genome-Wide Identification and Characterization of *SPL* Family Genes in *Chenopodium quinoa*

**DOI:** 10.3390/genes13081455

**Published:** 2022-08-16

**Authors:** Hongmei Zhao, Huaqi Cao, Mian Zhang, Sufang Deng, Tingting Li, Shuping Xing

**Affiliations:** 1College of Biological Sciences and Technology, Jinzhong University, Jinzhong 030600, Shanxi, China; 2College of Life Science, Shanxi University, Taiyuan 030006, Shanxi, China; 3Institute of Applied Biology, Shanxi University, Taiyuan 030006, Shanxi, China

**Keywords:** *Chenopodium quinoa*, *SPL* gene family, *Cqu*-*MIR156/7*, promoter analysis, expression pattern, stress response

## Abstract

*SQUAMOSA PROMOTER BINDING PROTEIN-LIKE* (*SPL*) genes encode a large family of plant-specific transcription factors that play important roles in plant growth, development, and stress responses. However, there is little information available on *SPL* genes in Chenopodiaceae. Here, 23 *SPL* genes were identified and characterized in the highly nutritious crop *Chenopodium quinoa*. Chromosome localization analysis indicated that the 23 *CqSPL* genes were unevenly distributed on 12 of 18 chromosomes. Two zinc finger-like structures and a nuclear location signal were present in the SBP domains of all CqSPLs, with the exception of CqSPL21/22. Phylogenetic analysis revealed that these genes were classified into eight groups (group I–VIII). The exon–intron structure and motif composition of the genes in each group were similar. Of the 23 *CqSPL*s, 13 were potential targets of miR156/7. In addition, 5 putative miR156-encoding loci and 13 putative miR157-encoding loci were predicted in the quinoa genome, and they were unevenly distributed on chromosome 1–4. The expression of several *Cqu-MIR156/7* loci was confirmed by reverse transcription polymerase chain reaction in seedlings. Many putative *cis*-elements associated with light, stress, and phytohormone responses were identified in the promoter regions of *CqSPL*s, suggesting that *CqSPL* genes are likely involved in the regulation of key developmental processes and stress responses. Expression analysis revealed highly diverse expression patterns of *CqSPL*s among tissues. Many *CqSPL*s were highly expressed in leaves, flowers, and seeds, and their expression levels were low in the roots, suggesting that *CqSPL*s play distinct roles in the development and growth of quinoa. The expression of 13 of 23 *CqSPL* genes responded to salt treatment (11 up-regulated and 2 down-regulated). A total of 22 of 23 *CqSPL* genes responded to drought stress (21 up-regulated and 1 down-regulated). Moreover, the expression of 14 *CqSPL* genes was significantly altered following cadmium treatment (3 up-regulated and 11 down-regulated). *CqSPL* genes are thus involved in quinoa responses to salt/drought and cadmium stresses. These findings provide new insights that will aid future studies of the biological functions of *CqSPL*s in *C. quinoa*.

## 1. Introduction

The expression of genes plays a critical role in the growth and development of plants, and transcription factors (TFs) are essential for regulating gene expression. TFs are classified into several families according to their sequences, structures, domains, motifs, and other molecular features. SQUAMOSA PROMOTER BINDING PROTEIN-LIKE (SPL) proteins are a family of plant-specific TFs that contain a conserved SBP domain of 76 amino acids. Two zinc finger sites are present in the N-terminal and C-terminal of the SBP domain, and the C-terminal zinc finger site overlaps with a nuclear localization signal (NLS) [1,2]. The first two SBP proteins, AmSBP1 and AmSBP2, were identified in snapdragon (*Antirrhinum majus*) based on their binding activity to the promoter region of the flower meristem identity gene *SQUAMOSA* [3]. Another two *SBP*-box genes, *SPL3* and *liguleless1* (*LG1*), were later described from *Arabidopsis* and maize, respectively. *SPL3* is involved in floral transition [4], and *LG1* is required for ligule and auricle formation during leaf development [5]. Additional *SBP*-box genes were later identified in *Arabidopsis* and rice [6,7]. A total of 4168 sequences encoding SBP proteins from green algae to land plants have been deposited in the database of plant TFs (http://planttfdb.gao-lab.org/, accessed on 2 March 2022). The *SPL* gene family has been analyzed at the genome-wide level in more than 60 species (Appendix A), including 15 *SPL* genes in grape, 28 in poplar, 31 in maize, 15 in potato, 24 in buckwheat, and 56 in wheat [8,9,10,11,12,13,14]. More *SPL* genes will likely be discovered as more plant genome resources become available.

*SPL8* was the first *SPL* gene to be functionally analyzed in *Arabidopsis* using a mutant. Three alleles of *spl8* mutants were obtained, and all evidence indicated that *SPL8* plays a role in microsporogenesis [15]. Overexpression analysis indicated that *SPL8* is involved in gibberellin signaling [16]. *SPL9* and *SPL15* have redundant functions in regulating shoot maturation and plastochron length [17,18]. Similar to the proteins AmSPB1 and AmSBP2 in snapdragon, SPL3 and SPL9 promote flowering by directly binding to the *cis*-elements of the flowering genes *LFY*, *FUL*, *AP1*, and *SOC1* [19,20]. In addition to the functions mentioned above, *SPL* genes play key roles in phase transitions [21,22], lateral root development [23], trichome formation [24,25,26], fertility [27,28], embryogenesis [29], and seedling growth [30]. In rice, *SPL* genes control grain size, shape, yield, and quality as well as plant architecture [31,32,33,34]. Rice yield can be enhanced by fine-tuning the expression of *SPL* genes [35]. In maize, *SPL* genes regulate bract development, primordium initiation, and domestication [36,37,38]. *SPL* genes regulate inflorescence morphogenesis, ovule development, and fruit ripening in tomato [39,40,41,42] and nodulation in soybean [43]. Recently, *TaSPL3/17* have been shown to be involved in tilling/branching in wheat [44].

*SPL* genes control copper (Cu) homeostasis and the response to Cu deficiency [45,46], cadmium (Cd) tolerance [47,48], and iron (Fe) homeostasis [49]; they are also involved in responses to other abiotic and biotic stresses, such as cold, heat, salt, drought, and wounding, as well as hormone signaling [50,51,52,53,54,55,56,57]. Many *SPL* genes are targeted by miR156/7; for example, 10 out of 16 *SPL* genes in *Arabidopsis* and 11 out of 19 from rice contain miR156/7 complementary sequences. This miR156/7-*SPL* module is a regulatory hub that links plant developmental stage transitions with other biological processes [58].

Quinoa (*Chenopodium quinoa* Willd.) is an annual grain-like crop that was originally grown in South Africa [59]. It began to be cultivated and domesticated over 7000 years ago [60]. The planting area of quinoa has significantly increased over the past decades, and it is now grown in North and South America, Europe, and Asia. Quinoa has received increased attention because of its highly nutritious seeds, which are rich in essential amino acids, minerals, and vitamins; it is also gluten-free and possesses a superior ratio of protein, lipids, and carbohydrates compared with other grains and grain-like crops [61]. In addition, quinoa shows high tolerance to environmental stresses, such as drought, salinity, and low temperature [62]. This, coupled with its low nutrient requirements, enables it to grow on arid soil and marginal land. Quinoa was declared by the United Nations Food and Agricultural Organization as a major crop for ensuring world food security and sustainability under global climate change, and 2013 was designated as the year of quinoa [63]. The first chromosome-scale reference genome of quinoa was published in early 2017 [64], and this genomic resource has provided key insights into the mechanisms underlying agronomically important traits of quinoa. The genes involved in some key biological processes have been studied [65,66]. Some TF families, such as NAC, GRAS, ARF, bZIP, and WRKY, have been studied at the genome level in quinoa [59,67,68,69,70]. However, *SPL* genes, which comprise a conserved gene family that might play a role in quinoa growth, development, and stress responses, have not yet been studied at the genome scale. Here, we identified *SPL* genes in quinoa on a genome-wide scale, determined the chromosomal locations of these genes, and analyzed the structure, phylogenetic relationships, conserved motifs, *cis*-acting elements, and expression profiles of these genes in various tissues under normal growth conditions. miR156/7 family genes were also identified, and their expression patterns in quinoa seedlings were examined. Putative targeted *SPL* genes were predicted. The expression responses of these quinoa *SPL* genes to salt, drought, and Cd stress were investigated to reveal their potential functions in stress tolerance. The results of this study provide new insights that enhance our understanding of the evolution and function of *CqSPL* genes. Our findings also contribute novel information that could be used to screen specific *CqSPL* genes involved in various developmental processes and the response to different types of abiotic stress.

## 2. Materials and Methods

### 2.1. Plant Materials, Growth Conditions, and Stress Treatments

Quinoa plants were grown in a growth chamber under controlled conditions: 60% relative humidity, 418.2 ppm CO_2_, a 16 h/8 h light/dark cycle, and temperature of 23 ± 1 °C. The light was produced from three band fluorescent lamps (T5/28W), and the light intensity was 120 µmol/m^2^/s. The plant samples were taken from two-week-old seedlings and seeds produced after flowering.

To determine the expression of *CqSPL* genes under stress conditions, quinoa seeds were germinated on moist filter paper in a growth chamber. Three-day-old uniform seedlings were transplanted into the holes of a 96-hole plate in a 2-L container filled with 1/2 Hoagland nutrient solution. On the seventh day after transplanting, the seedlings were subjected to different treatments: (1) control—seedlings were kept in nutrient solution throughout the experiment; (2) drought stress—seedlings were transferred to nutrient solution containing 15% PEG6000; (3) salt stress—seedlings were transferred to nutrient solution with 200 mM NaCl; and (4) Cd stress—seedlings were transferred to nutrient solution with 100 μM CdCl_2_. Samples were collected 2 h after each treatment and stored at −80 °C until analysis.

### 2.2. Identification of SPL and MIR156/7 Family Members in the Quinoa Genome

*SPL8* from *Arabidopsis thaliana* (*AtSPL8*) was used as a query to conduct BLAST searches against the Chemopodium DB (http://www.cbrc.kaust.edu.sa/chenopodiumdb/, accessed on 25 February 2022) and Phytozome 13 (http://www.phytozome-next.jgi.doe.gov/, accessed on 25 February 2022) to identify *SPL* genes in the quinoa genome. Twenty-three sequences homologous to *AtSPL8* were recovered in the BLAST results, and each sequence was further confirmed using SMART ((http://smart.embl-heidelberg.de/, accessed on 26 February 2022) and CDD ((http://www.ncbi.nlm.nih.gov/Structure/cdd/wrpsb.cgi/, accessed on 26 February 2022)) from the National Center for Biotechnology Information (NCBI) based on the presence of the SBP domain. Other *SPL*s from *A. thaliana* were also used as queries to conduct BLAST searches, but no additional sequences were recovered. The molecular weight (MW) and isoelectric point (pI) were predicted using a EXPASy Proteomics Server (https://web.expasy.org/compute_pi/, accessed on 5 March 2022). Subcellular localization prediction was performed using Plant-mSubP (http://www.bioinfo.usu.edu/Plant-mSubP/, accessed on 5 March 2022).

To search for *MIR156/7* genes, precursor sequences of *MIR156/7* from *A. thaliana* were used to conduct BLAST searches against the Chemopodium DB, and the secondary structures of the obtained sequences were predicted using MFold (http://www.mfold.org/, accessed on 20 March 2022)*. Cqu-MIR156* family members in PmiREN (http://www.pmiren.com/, accessed on 25 March 2022) were individually compared with sequences obtained from our BLAST searches. All web softwares mentioned above were used with default parameters.

### 2.3. Location of Genes on Chromosomes

The physical map of the chromosomes of quinoa was scaled following Jarvis et al. [64]. The distribution of *CqSPL* and *Cqu-MIR156/7* genes in the quinoa genome was determined using the ChenopodiumDB V 1.0 (V1 pseudomolecule) annotation database (https://www.cbrc.kaust.edu.sa/chenopodiumdb/, accessed on 4 April 2022). The chromosome location map was built using MG2C (mg2c.iask.in/mg2c_v2.1/, accessed on 4 April 2022).

### 2.4. Prediction of CqSPL Genes Targeted by miR156/7

Based on information from *Arabidopsis*, most sequences targeted by miR156/7 are present in the coding region of the target genes, and this conserved region encodes six conserved amino acids, ALSLLS [27]. A search of the conserved ALSLLS sequence in 23 CqSPLs revealed that 12 CqSPLs contained this conserved amino acid sequence, suggesting that these 12 *CqSPLs* are putative targets of Cqu-miR156/7. The full-length genomic sequences of 23 *CqSPL* genes were analyzed using the psRNATarget online tool (http://plantgrn.noble.org/psRNATarget/, accessed on 25 April 2022) with default parameters. We detected another gene with this same target element in the 3′ UTR. Multiple sequence alignment of the miR156/7-targeted *CqSPL* genes and reverse-complement sequences of Cqu-miR156/7 was performed using the ClustalW method in MEGA11. This same method was used to compare the miR156/7 sequences from both *A. thaliana* and *C. quinoa*.

### 2.5. Sequence Alignment and Phylogenetic Analysis

*AtSPL* sequences were obtained from the TAIR home page (http://www.arabidop sis.org/, accessed on 10 February 2022). *CqSPL* sequences were downloaded from Phytozome (http://www.phytozome-next.jgi.doe.gov/, accessed on 10 February 2022). The phylogenetic tree was generated using ClustalW alignment and the neighbor-joining method in MEGA 11 with 1000 bootstrap replicates and provided settings (Jones–Taylor–Thornton model, uniform rates, pairwise deletion, and number of threads 7). The orthologship of the genes in the phylogenetic tree was confirmed as reciprocal best hits using BLAST in NCBI.

### 2.6. Gene Structure and Conserved Motifs

To clarify the structures of *CqSPL* genes, genomic sequences and coding sequences (CDSs) were downloaded from Phytozome and the Chenopodium DB. Graphs of exon–intron structures were obtained using GSDS 2.0 (gsds.gao-lab.org/, accessed on 13 April 2022) by inputting genomic sequences and related CDSs. The MEME online tool (http://meme-suite.org/tools/meme/, accessed on 20 April 2022) was used to analyze the full-length CqSPL proteins to predict the conserved motifs using the following parameters: motif width, 10 to 50; maximum number of conserved motifs, 20; and site distribution, zero or one occurrence per sequence. The SBP-domain sequence logo was prepared using the WebLogo online tool (weblogo.berkley.edu/logo.cgi/, accessed on 25 April 2022).

### 2.7. Identification and Analysis of Promoters

The 3.0-kb promoter sequences of the 23 *CqSPL* genes were retrieved from the Che nopodium DB. MEME (http://meme-suite.org/tools/meme/, accessed on 15 April 2022) was used to predict the conserved motifs; the maximum number of motifs was set to 30 with a motif width of 6 to 50. Zero or one occurrence per sequence was chosen as the site distribution. The promoters were also analyzed using PlantCARE software (http://bioinformatics.psb.ugent.be/webtools/plantcare/html/search_CARE.html/, accessed on 20 April 2022) and the New PLACE database (https://www.dna.affrc.go.jp/PLACE/, accessed on 20 April 2022) to identify *cis*-regulatory sequences [71].

### 2.8. RNA Extraction and qRT-PCR Analysis

Total RNA was extracted from harvested samples using an RNeasy Plant Mini Kit (Qiagen), following the manufacturer’s instructions. The RNA quality and concentration were determined using a NanoDrop 2000 spectrophotometer (Thermo Fisher Scientific, Waltham, MA, USA), and 2.0 μg of total RNA was used for first-strand cDNA synthesis with a reverse-transcription kit (Takara Bio Group, Shiga, Japan). Quantitative real-time polymerase chain reaction (qRT-PCR) was performed using a BIO-RAD CFX Connect™ Real-Time System with ChamQTM Universal SYBR qPCR Master Mix (Vazyme, Nanjing, China). The primer sequences used are listed in Appendix A. The total volume of the qRT-PCR reaction was 20 μL, and it contained 10 μL of 2× SYBR qPCR Master Mix, 0.8 μL of primers (10 μM), 2 μL of cDNA templates, and 6.4 μL of ddH_2_O. The reaction was conducted under the following thermal cycling conditions: 94 °C for 2 min, followed by 40 cycles of 94 °C for 15 s, and 60 °C for 31 s. *CqACT2* was used as an internal control. The experiments were conducted in three biological replicates. The relative expression levels of genes were calculated using the 2^−ΔΔCT^ method.

### 2.9. Data Analysis

Statistical analyses were carried out by one-way ANOVA, followed by Tukey’s HSD test with SPSS v16.0. All data are shown as mean ± SE, and statistical significance was considered at *p* < 0.05 (* *p* < 0.05, ** *p* < 0.01, *** *p* < 0.001).

## 3. Results

### 3.1. Identification of SPL Family Genes in the Quinoa Genome

In the quinoa genome, 23 *SPL* genes were retrieved using BLASTP. Two orthologous genes of *AtSPL8* were obtained and designated as *CqSPL8A* and *CqSPL8B*. Other genes were named *CqSPL1* to *CqSPL22* based on their order of appearance in the BLAST results. *CqSPL8A* was located on chromosome 1, and *CqSPL8B* was located on chromosome 10. Other *CqSPLs* were unevenly distributed on chromosome 3, 6–12, and 14–16 (Figure 1). The predicted CDSs of the *CqSPL* genes ranged between 609 and 3570 bp, and the deduced length of the proteins ranged from 202 to 1189 amino acids (Table 1, Appendix A). A short version of the two *CqSPL8* genes (*CqSPL8A* and *CqSPL8B*) was initially identified in the BLAST search, with a 570-bp CDS and a deduced protein of 190 amino acids. Comparison with *SPL8* orthologs from other species such as *Beta vulgaris* and *Solanum lycopersicum*, which have SPL8 proteins with more than 280 amino acids, provided confirmation of the full-length *CqSPL8A* and *CqSPL8B* sequences, which have 849-bp and 861-bp CDSs, respectively (Table 1). The predicted MW of these CqSPLs ranged from 21.12 to 132.14 kD, and the pI that ranged from 5.74 to 9.81. CqSPL3, CqSPL4, and CqSPL5 were predicted to be localized to both the nucleus and cytoplasm; the rest of the CqSPLs were exclusively localized to the nucleus (Table 1).

In addition, consistently with findings in other plant species, we found that 13 of 23 *CqSPL* genes contained miR156/7-targeted elements, and all were located in the coding regions, with the exception of *CqSPL6*, which had the mir156/7-targeted sequence in its 3′UTR (Figure 2A).

Accordingly, 18 *MIR156/7* genes in the quinoa genome were revealed (Appendix A), including 16 that had previously been added to the PmiREN database (https://pmiren.com/, accessed on 2 April 2022). We renamed these *MIR156/7* genes based on the mature sequences of miR156/7; miR157 subfamily sequences contain a U at the tenth position from the 3′ end, and miR156 subfamily sequences contain a U/A at the seventh position from the 3′ end (Figure 3). Five *MIR156* genes were named *Cqu-miR156a* to *Cqu-miR156e*. The other 13 were *MIR157* subfamily genes and were named *Cqu-miR157a* to *Cqu-miR157m* (Table 2). Eleven *Cqu-MIR156/7* genes were located on chromosome 4, and the *Cqu-MIR157a-d* cluster and *Cqu-MIR157e-h* cluster are duplicates of each other. *Cqu-MIR156a* and *Cqu-MIR157l* were located near both ends of chromosome 4, and *Cqu-MIR156b* was located below the *Cqu-MIR157e-h* cluster. *Cqu-MIR156e* and *Cqu-MIR156c* were located on chromosome 2 and 3. *Cqu-MIR156d*, *Cqu-MIR157i-k*, and *Cqu-MIR157m* were located on chromosome 1 (Figure 1). *Cqu-MIR156e* and *Cqu-MIR157m* were newly discovered in this study. The precursor sequences of these two miRNAs can both form a hairpin structure with the mature sequence of miR156/7 on the stem (Figure 2B).

We examined the expression of several *Cqu-MIR156/7* loci in quinoa seedlings. All the *Cqu-MIR156/7* genes examined were expressed in the seedlings (Appendix A).

### 3.2. Phylogeny of CqSPL Genes

To explore the evolutionary relationship between the *SPL* genes from both quinoa and *A. thaliana*, a phylogenetic tree of 23 *CqSPL*s and 17 *AtSPL*s was constructed based on the conserved SBP-domain sequences (Figure 4). The tree can be divided into eight groups, with at least one gene from both species in each group. Group I contained four genes (*CqSPL12*, *CqSPL13*, *CqSPL19*, and *CqSPL20*) from *C. quinoa* and only one gene (*AtSPL6*) from *A. thaliana*. Group II had four genes: two (*CqSPL7* and *CqSPL9*) from *C. quinoa* and two (*AtSPL9* and *AtSPL15*) from *A. thaliana*. Three genes (*AtSPL11*, *AtSPL10*, and *AtSPL2*) from *A. thaliana* and two genes (*CqSPL1* and *CqSPL2*) from *C. quinoa* were present in Group III. Two *SPL13* duplicates (*AtSPL13A* and *AtSPL13B*) from *A. thaliana* and four *CqSPL* genes (*CqSPL17*, *CqSPL18*, *CqSPL11*, and *CqSP14*) from *C. quinoa* were present in Group IV. Group V comprised *SPL8* orthologs (*CqSPL8A*, *CqSPL8B*, and *AtSPL8*). There were ten genes in group VI: six (*CqSPL3*, *CqSPL4*, *CqSPL5*, *CqSPL10*, *CqSPL15*, and *CqSPL16*) from *C. quinoa* and four (*AtSPL1*, *AtSPL12*, *AtSPL14*, and *AtSPL16*) from *A. thaliana*. Group VII contained four genes: three genes (*AtSPL3*, *AtSPL4*, and *AtSPL5*) from *A. thaliana* and one gene (*CqSPL6*) from *C. quinoa*. Group VIII comprised three genes: two (*CqSPL21* and *CqSPL22*) from *C. quinoa* and one (*AtSPL7*) from *A. thaliana*. The genes in group I–IV and group VII were targets of miR156/7 in both species (Figure 2A), and the genes in the other groups were not targets of miR156/7. Ten genes from group VII encoded large proteins consisting of more than 800 amino acids. *AtSPL7* in group VIII encoded a protein of 818 amino acids, but the orthologs *CqSPL21* and *CqSPL22* in this group encoded proteins of approximately 710 amino acids. The length of the other SPLs from both species was less than 600 amino acids. Phylogenetic analysis revealed that every group of *CqSPL* genes had at least one ortholog in *A. thaliana*.

### 3.3. Structure of CqSPL Genes and Domain/Motif Analysis of Their Proteins

To compare the 23 *CqSPL* genes directly, their exon–intron structures were first predicted. In general, the number of introns within the coding regions of *CqSPL* genes ranged from 2 to 14. Eight genes (*CqSPL6*, *CqSPL8A*, *CqSPL8B*, *CqSPL11*, *CqSPL12*, *CqSPL13*, *CqSPL14*, and *CqSPL20*) had two introns, and two genes (*CqSPL4* and *CqSPL15*) had fourteen introns. The largest intron was intron 2 (8584 bp), which was present in *CqSPL19*, and the smallest intron was intron 5 (75 bp), which was present in *CqSPL16*. The widest exon was present in *CqSPL12* (3854 bp), and the shortest exon was present in *CqSPL15* (exon 9, 24 bp). *SBP*-box sequences in most *CqSPLs* spanned two exons, but they were present in only one exon in a subset of large *CqSPLs* (*CqSPL3*, *CqSPL4*, and *CqSPL5* in group VI). One *SBP*-box sequence in *CqSPL6* spanned three exons (Figure 5). Several pairs of *CqSPL* genes were identified, including *CqSPL1/2*, *CqSPL8A/B*, *CqSPL11/14*, and *CqSPL12/13*. Each pair of *CqSPL* genes had the same number of introns with the same intron phase and were in the same group (Figure 4).

The domain conserved among all CqSPLs was the SBP domain, which comprised 76 amino acids; in CqSPL21 and CqSPL22, several parts of the SBP domain were missing (Figure 6A). As in other species, the SBP domain of CqSPLs contained two zinc finger structures: one (Zn1, CCCH) in the N-terminal and another (Zn2, CCHC) in the C-terminal. The conserved bipartite NLS overlapped with the second zinc finger structure (Zn2) (Figure 6A,B). The CqSPL21 had 17 fewer amino acids around the Zn2 motif, and 32 amino acids were absent around both the Zn1 and Zn2 motifs in CqSPL22 (Figure 6A).

Additional motifs were identified in CqSPLs using MEME. Twenty motifs were identified from all CqSPLs, and the SBP-box domain was present in the connected motifs 3, 1, and 4. Of the 20 motifs, 18 were identified in the 3 large proteins CqSPL3, 4, and 5. CqSPL10 and CqSPL16 both had 13 motifs, and CqSPL15 had 11 motifs. No additional motifs were detected in CqSPL6 aside from motifs 3, 4, and 1 (SBP domain). There was only one motif (motif 8) that overlapped with the N-terminal part of the SPL8-specific motif (RIGLNLGGRTYF from AtSPL8) upstream of the SBP domain in CqSPL8A and CqSPL8B. The motif composition was similar among CqSPLs from the same group; for example, CqSPL1 and CqSPL2 in Group III (Figure 4) each had an upstream motif (motif 15) and downstream motif (motif 11 containing the miR156/7 target site) with the SBP domain, as well as the connected motifs 3, 1, and 4 containing the SBP-box domain. CqSPL21 and 22 from group VIII had 3 motifs (motifs 2, 9, and 10) downstream of their partial SBP domains (motifs 3 and 4, Figure 7).

### 3.4. Promoter Analysis of CqSPL Genes

*Cis*-acting elements are required for the proper expression of genes. We studied the motifs and *cis*-elements in the 3.0-kb promoters of 23 *CqSPL* genes (Appendix A). We first generated a phylogenetic tree of these promoters (Figure 8A). Promoters from gene pairs or genes from the same group in Figure 4 were generally clustered (Figure 8A). Multiple motifs were determined in both strands of the *CqSPL* promoters using MEME. The number of the motifs ranged from 7 (*CqSPL20*) to 23 (*CqSPL14*). Of the 23 *CqSPL* promoters, 17 had 10 or more motifs. The spatial distribution of these motifs was similar within promoter groups but differed among promoter groups. For example, the *CqSPL3*/*4*/*5* group contained 12 or 16 motifs in their promoters, and most of these motifs were located within 1 kb upstream (−1 to 0 kb) of the start codon, and motif 22 and 23 in the *CqSPL11*/*14* group occurred further upstream (−3 to −1 kb) of the promoters (Figure 8A,B).

To obtain detailed information on the *cis*-elements or motifs in the *CqSPL* promoters, PlantCARE software was used to analyze promoter sequences. A total of 94 types of *cis*-elements were identified, and they were unevenly distributed in the 3.0-kb upstream region of the *CqSPL* genes. The copy number of these *cis*-elements ranged from 171 in *CqSPL22* to 272 in *CqSPL8B*. The basic *cis*-elements TATA-box and CAAT-box were the most common in these motifs; for example, 115 and 81 copies of TATA-box and CAAT-box were present in *CqSPL8A*, and 42 TATA-boxes and 54 CAAT-boxes were present in *CqSPL16*. Other *cis*-elements were classified into several categories: light responsiveness, development, hormone responsiveness, stress responsiveness, elicitor induction, and unknown function (Appendix A). *Cis*-elements involved in stress responsiveness, including abiotic and biotic stress, were detected in all 23 *CqSPL* promoters and were further validated in New PLACE database. MYB and MYC elements are drought-responsive and dehydration-responsive elements, respectively, and both were present in the 23 *CqSPL* promoters (Appendix A). The copy numbers of these *cis*-acting elements in each promoter are shown in Figure 9. All these promoters contained more than 20 copies of these elements, with the exception of *CqSPL8A*, which had only 12 copies. A maximum of 43 copies of these elements were detected in *CqSPL7*, and 40 were present in both *CqSPL9* and *CqSPL10* (Figure 9 and Appendix A). These data suggest that *CqSPL* genes play a role in developmental processes and stress responses.

### 3.5. Tissue Expression Patterns of CqSPL Genes

To have a general idea where the *CqSPL* genes function, the tissue expression patterns of the *CqSPL* genes were determined by performing qRT-PCR on samples taken from several tissues, including root, seedling, stem, leaf, flower, and seed tissue. The expression level of most *CqSPL* genes was lower in the roots and higher in flowers and leaves. The expression of some *CqSPL* genes was higher in flowers compared with other tissues, such as *CqSPL6*, *CqSPL8A*, *CqSPL8B*, *CqSPL11*, and *CqSPL14*. *CqSPL21* was most highly expressed in the leaves, *CqSPL4* and *CqSPL17* were most highly expressed in seeds, and *CqSPL18* was most highly expressed in seedlings. Variation in the expression levels of the other genes among tissues was low. Genes with similar motif compositions in their promoters had similar expression patterns (Figure 8 and Figure 10). For example, *CqSPL1/2*, *CqSPL8A*/*8B*, *CqSPL11/14*, and *CqSPL15*/*16* were all from the same group and had similar motif compositions and expression patterns (Figure 4 and Figure 8). However, this was not the case for the *CqSPL3*/*4*/*5* group, as *CqSPL4* was highly expressed in flowers and seeds, *CqSPL3* was highly expressed in leaves, and *CpSPL5* was highly expressed in flowers (Figure 10). These variable expression patterns indicate that the *CqSPLs* involved in development and other physiological processes are functionally diverse.

### 3.6. Expression of CqSPL Genes under Drought, Salt, and Cd Stress

To evaluate the functions of the *CqSPL* genes in response to various types of stress, qRT-PCR was performed after quinoa seedlings were exposed to salt (NaCl), drought (PEG), and Cd treatment. Comparison of the expression level between the control and stress-treated seedlings revealed that the expression of 13, 22, and 14 *CqSPL* genes significantly responded to salt, drought, and Cd treatment, respectively. In the salt treatment, the expression of 11 genes was up-regulated, and the expression of 2 genes was down-regulated. Of the 11 up-regulated genes, the expression of *CqSPL8B*, *CqSPL11*, and *CqSPL18* was up-regulated more than 2.5, 2.2, and 8.2-fold, respectively, and the expression of the two down-regulated genes *CqSPL6* and *CqSPL19* was decreased more than 9.7 and 2.4-fold, respectively (Figure 11A). In the drought stress treatment, the expression of 21 *CqSPL* genes was up-regulated, and the expression of *CqSPL3*, *CqSPL4*, *CqSPL5*, *CqSPL12*, and *CqSPL18* was increased more than 15.1, 14.0, 30.6, 13.2, and 27.0-fold, respectively. The expression of all other up-regulated genes, with the exception of *CqSPL1* and *CqSPL2*, was increased by at least more than 2-fold. The expression of *CqSPL6* was down-regulated more than 2.4-fold in the drought treatment (Figure 11B). The expression of only three *CqSPL* genes (*CqSPL4*, *CqSPL17*, and *CqSPL18*) was up-regulated in the Cd treatment, and the expression of these three genes was up-regulated 1.6, 1.9, and 10.1-fold, respectively. The expression of 11 *CqSPL* genes was down-regulated; the expression of *CqSPL6* and *CqSPL22* was decreased more than 4.5 and 2.6-fold, respectively, and the expression of the other 9 down-regulated *CqSPL* genes was decreased by less than 2-fold (Figure 11C). Generally, the expression of *CqSPL4*, *CqSPL18*, and *CqSPL6* responded to all three types of stress and exhibited similar expression patterns; the expression of *CpSPL4* and *CqSPL18* was up-regulated, and the expression of *CqSPL6* was down-regulated. These findings indicated that many *CqSPL* genes are involved in the response to two types of stress.

## 4. Discussion

Since the first two *SBP*-box genes were identified in snapdragon [3], the number of *SPL* genes identified has significantly increased in many plant species. The SBP-box family is now a large plant gene family. *SPL* genes have only been detected in green plants to date, including green algae and land plants. Phylogenetic analysis has shown that the *SPL* genes of green algae and land plants form a monophyletic group and that each lineage of *SPL* genes has undergone duplication events followed by divergence; this suggests that *SPL* genes might have originated from a common ancestor of green plants and that the origin of *SPL* genes might predate the divergence between green algae and the ancestors of land plants [72,73]. Analysis of CqSPL proteins using SMART (http://smart.embl-heidelberg.de/, accessed on 4 March 2022) and grouping of proteins with similar domain architecture recovered a sequence with an SBP domain comprising 84 amino acid residues in the bacterium TMED181, and this sequence was similar to a sequence in green algae *Micromonas commode* (Appendix A). This potentially suggests that *SPL* genes might also occur in bacteria. However, additional work is needed to confirm this finding, given that this sequence might be derived from the contamination of bacterial samples. Thus, the origin of the *SPL* genes still remains ambiguous.

In this study, a total of 23 *CqSPL* genes were identified in the quinoa genome, and these genes were clustered into 8 groups with 17 members from *Arabidopsis* (Figure 4). Similar groupings of *SPL* genes have been reported in petunia, tomato, tartary buckwheat, sugarcane, and *Jatropha curcas*; however, the group order of these genes might vary among species [13,74,75,76,77]. These data suggest that *CqSPL* genes might have originated from their close relatives. Quinoa is an allotetraploid plant (2n = 4x = 36) that might be derived from a cross between two diploids such as *Chenopodium pallidiaule* (2n = 2x = 18, A sub-genome) and *Chenopodium suecicum* (2n = 2x = 18, B sub-genome). We identified 10 and 12 *SPL* genes in *C. palidiaule* and *C. suecicum* from the Phytozome database (https://phytozome-next.jgi.doe.gov/, accessed on 25 May 2022), respectively (Appendix A), and this might provide insight into the distribution and divergence of *CqSPL* genes in the quinoa genome.

Notably, two studies examining the evolution of *SPL* genes have suggested that *SPL* genes from land plants can generally be divided into two distinct groups. Group I members generally occur in single copies or a low number of copies in each species; group II contains several members in each species and can be further classified into several subgroups [72,73]. One marked difference between group I and II members is that the N-terminal zinc finger of the SBP domain is C4 (CysCysCysCys) in group I but C3H (CysCysCysHis) in group II (the C-terminal zinc finger is C2HC in both group I and II). AtSPL7 from *Arabidopsis* belongs to group I. It plays an important role in Cu homeostasis [45] and is functionally related to CRR1 in the single-celled green alga *Chlamydomonas reinhardtii* [78], but CRR1 possesses a C3H zinc-finger in the N-terminal of the SBP domain, similar to other SPL members in green algae (Appendix A) and group II members in land plants. In our phylogenetic analysis, AtSPL7, CqSPL21, and CqSPL22 were classified into group VIII (Figure 4). Although CqSPL21 and CqSPL22 only contained a partial SBP domain, CqSPL21 still had a full N-terminal zinc finger C4 (Figure 6). A deletion of the same region within the SBP domain of CqSPL22 has also been observed in *C. pallidiaule* (A sub-genome contributor), but this was not the case for CqSPL21, as its ortholog in *C. suecicum* (B sub-genome contributor) has the full-length SBP domain. No deletion has been observed in other members of this group, including orthologs from *Beta vulgaris* and *Spinacia oleracea* (Appendix A), two species from the same family of quinoa (Chenopodiaceae). These findings indicate that the deletion of 32 amino acids in the SBP domain of CqSPL22 predated the speciation of quinoa, and the deletion of 17 amino acids in the SBP domain of CqSPL21 occurred following the speciation of quinoa. Whether these deletions affect the DNA-binding activity of CqSPL21 and CqSPL22, and whether these genes play a role in Cu homeostasis, requires further investigation. The rest of the groups (I to VII) in our study corresponded to the several subgroups (IIa to IIf) of group II in Guo et al. [72]. Group V (IIb) contained AtSPL8, CqSPL8A, and CqSPL8B (or the orthologs from other species). Recently, *SPL8* in alfalfa has been shown to play a role in biomass yield and the response to salt and drought stress [79]. *SPL8* has also been shown to promote flowering in switchgrass [80]. There has been much interest in studying the function of *SPL8,* but no study has yet addressed whether *SPL8* has a conserved function in early anther development and sporogenesis [15]. Members of group I–IV (IIc, IIe, and IIf) and group VII (IId) targeted by miR156/157 (or miR529) have been identified in land plants [72]. miR156/529 sequences have been identified in the red alga *Eucheuma denticulatum* [81], and this finding has raised questions regarding the origin of these miRNAs. These miR156/7-targeted *SPL* genes also show more diverse tissue-specific expression patterns (Figure 10) [82], which is consistent with the finding that this miR156/7-SPL module plays a key role in many developmental processes [58,82]. These miR156/7-targeted *SPL* genes encode smaller proteins than the *SPL* genes in group VI (IIa), which are not targeted by miR156/7. *AtSPL14*, *AtSPL16*, and *AtSPL1/12* from *Arabidopsis* in group VI are ubiquitously expressed in most tissues [82]. AtSPL14 and AtSPL1/12 have been shown to play a role in toxin resistance and thermotolerance, respectively [83,84]. However, the function of AtSPL16 has not yet been identified. Gene structure analysis has revealed that there are ten or more exons in the members of this group, with the exception of *ppSBP2* and *ppSBP10* from moss [72,82,85]. The six members in this group from quinoa (*CqSPL3/4/5*, *CqSPL10*, and *CqSPL15/16*) were highly expressed in most tissues examined, with the exception of root tissue (Figure 10). The structure of *CqSPL3/5/10/16* was similar to that of *AtSPL1/12/14*, and *CqSPL4/15* have more exons in their C-terminal region compared with other *CqSPL* genes (Figure 5). Another distinctive feature of *CqSPL3/4/5* was that the SBP domain-encoding sequence was only present in one exon, rather than two as in most other *SPL* genes. This pattern has also been observed in moss *ppSBP2* and *ppSBP10*, *AAA-12591* (*C. pallidiaule*), *BBB-13790* and *BBB-14488* (*C. suecicum*), *Spov_chr3.03860* (*S. oleracea*), and *EL10Ac3g06264* (*B. vulgaris*), which all belong to the same group. Whether this stems from an intron loss in these *SPL* genes or an intron gain in other *SPL* genes remains unclear.

Finally, the promoter of each *CqSPL* gene was analyzed to gain further insight into their functions. Many *cis*-elements involved in stress responses were identified in these *SPL* genes (Figure 9), suggesting that they play a key role in stress responses. Quinoa is a salt and drought-tolerant plant, but the mechanisms underlying its salt and drought tolerance remain unclear. miR156 and *SPL* genes have recently been reported to be involved in salt and drought responses [51,54,55,56,86,87,88,89,90]. miR156 and *SPL* genes have also been shown to play a role in responses to heavy metal stress, such as Cd and Fe stress [48,49]. Our qRT-PCR data revealed that the expression of many *CqSPL* genes responded to salt, drought, and Cd treatment (Figure 11), which suggests that these *CqSPL* genes are involved in these stress responses. Additional studies are needed to clarify the regulatory roles of these *CqSPL* genes in the response to salt, drought, and Cd stress in quinoa.

## 5. Conclusions

We identified and characterized 23 *SPL* genes in the quinoa genome, and these *SPL* genes could be divided into 8 groups with similar exon–intron structures and motif compositions. Of 23 *CqSPL* genes, 13 were putative targets of miR156/7. A total of 18 *MIR156/7* loci were predicted, and the expression of several of these genes was further confirmed in seedlings and seeds. Many *cis*-acting elements involved in light, hormone, and stress were identified in the promoter regions of these *CqSPL* genes. The diverse expression patterns of *CqSPL* genes among tissues and in response to salt, drought, and Cd stress suggest that these *CqSPL* genes play an important role in the growth, development, and stress responses of quinoa. The findings of this study provide new insights into the molecular mechanisms by which *CqSPL* genes regulate developmental and physiological processes, as well as information that will aid future studies of *SPL* genes in the Chenopodiaceae family.

## Figures and Tables

**Figure 1 genes-13-01455-f001:**
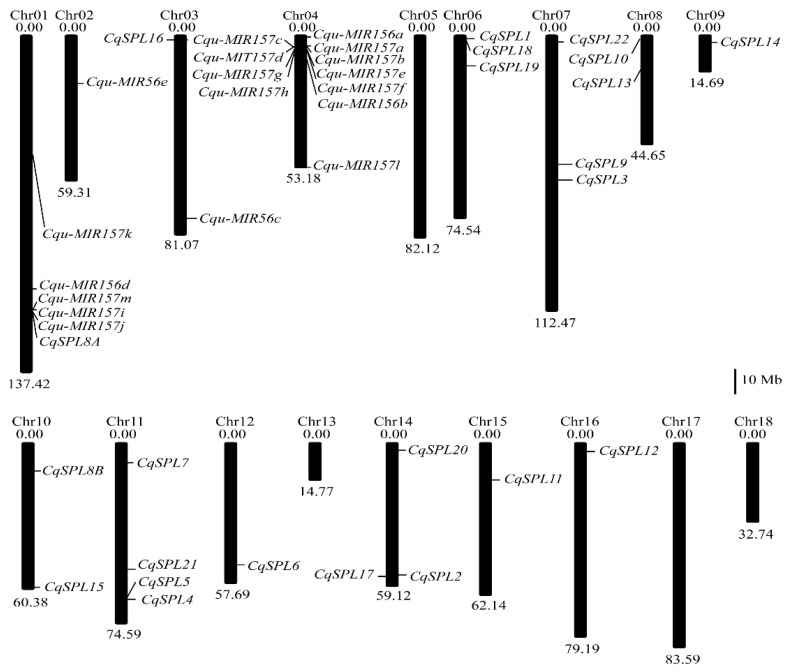
Chromosomal locations of *CqSPL* and *Cqu-MIR156/7* genes. The scale bar represents 10 megabases (Mb).

**Figure 2 genes-13-01455-f002:**
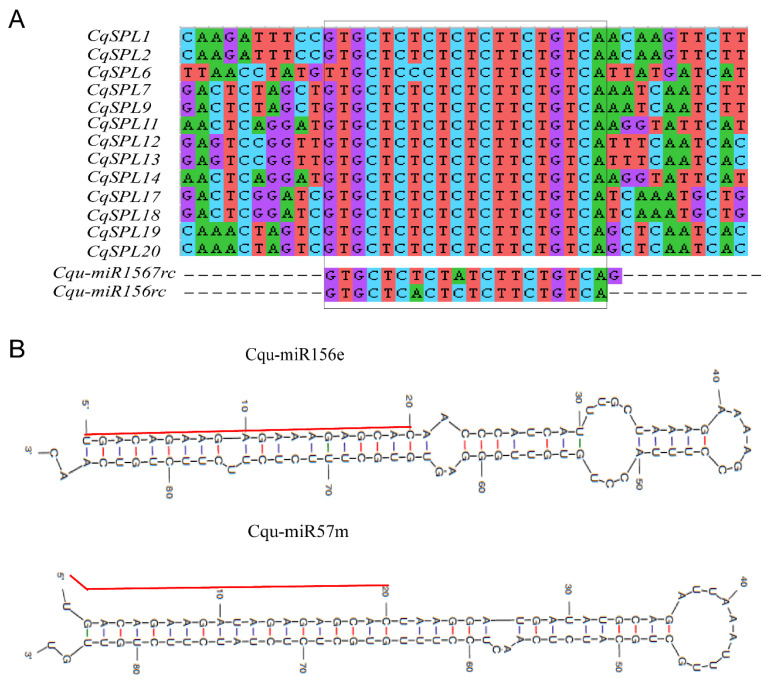
Sequence alignment of miR156/7 complementary sequences within *CqSPL* genes and predicted stem-loop hairpin structures. (**A**) Sequence alignment of miR156/7-targeted elements located in the coding regions of *CqSPL*s, with the exception of *CqSPL6*, which has the miR156/7-targeted sequence in its 3′ UTR. Reverse complementary sequences of the mature miR156/7 are shown below the alignment for comparison. (**B**) Hairpin structures predicted for Cqu-miR156e and Cqu-miR157m by MFold (http://www.mfold.org/, accessed on 5 April 2022). The mature sequences of miR156e and mi157m on the stems are indicated by red lines.

**Figure 3 genes-13-01455-f003:**
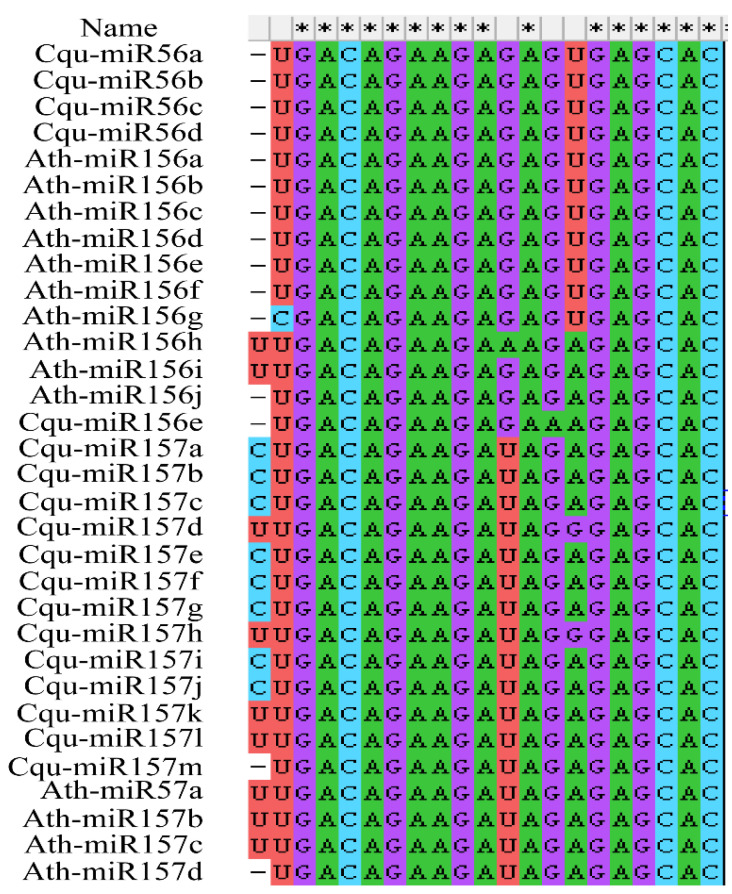
Sequence alignment of mature Cqu-miR156/7 and Ath-miR156/7. A U/A is present at the seventh position from the 3′ end of miR156 subfamily members, and a U is present at the tenth position from the 3′ end of miR157 subfamily members. The asterisk indicates the conserved nucleotide.

**Figure 4 genes-13-01455-f004:**
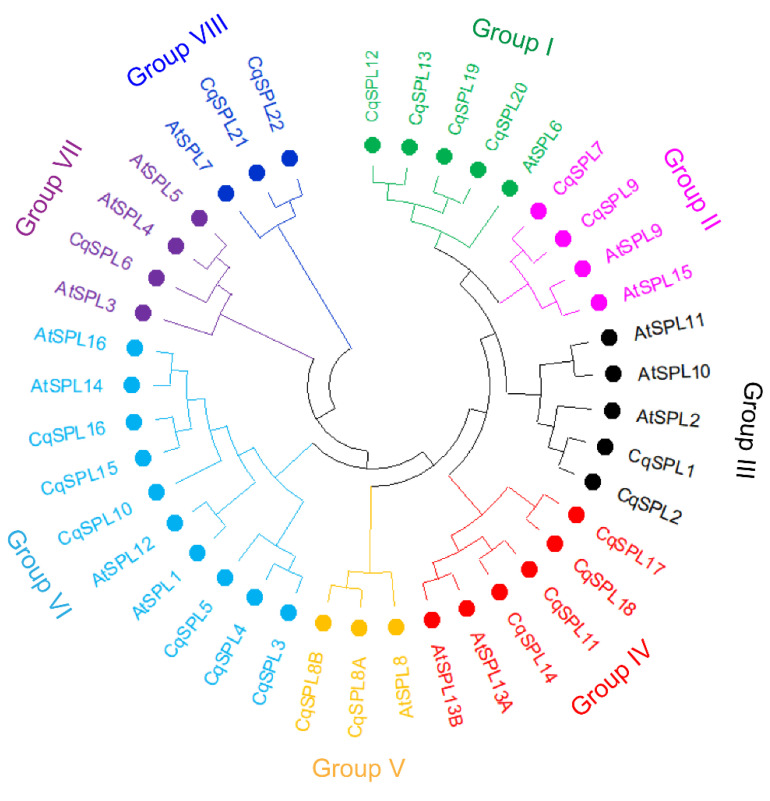
Phylogenetic analysis of SPLs from *C. quinoa* and *A. thaliana*. The phylogenetic tree was constructed based on the SBP-box domain sequences with MEGA 11 using the neighbor-joining method and 1000 bootstrap replicates. The defined groups were labeled with different colors and filled circles.

**Figure 5 genes-13-01455-f005:**
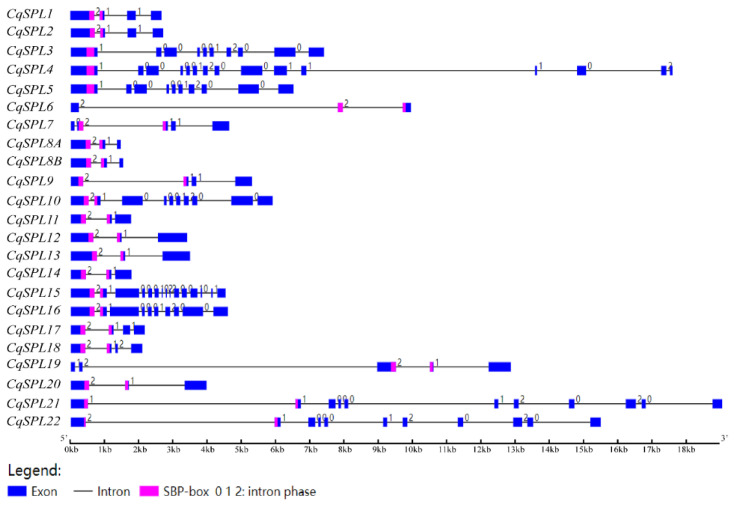
Gene structure of *SPL* family members in *C. quinoa*. The exons and introns are indicated by blue boxes and black horizontal lines, respectively. The pink color indicates the SBP box, and the numbers refer to the intron phase.

**Figure 6 genes-13-01455-f006:**
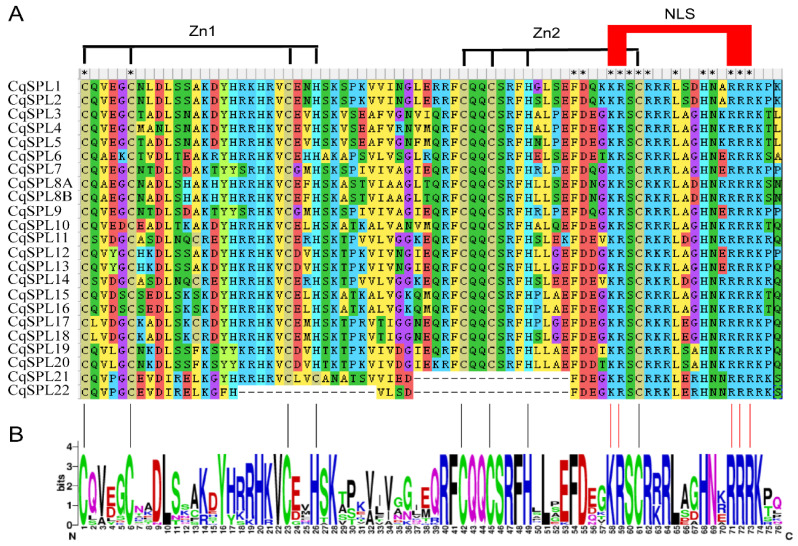
SBP-box domain alignment of CqSPLs. (**A**) Multiple alignment of SBP-box domains of CqSPL proteins. All CqSPL proteins contain an SBP-box domain of 76 amino acids, with the exception of CqSPL21 and CqSPL22, which have some missing parts of the SBP-box domain. The two conserved zinc finger motifs (Zn1 and Zn2) and the nuclear localization signal (NLS) are indicated. The asterisk indicates the conserved nucleotide. (**B**) The sequence logo of SBP-box domains (with the exception of that from CqSPL21 and CqSPL22) was prepared using the WebLogo online tool. The overall height of the stack reflects the extent of sequence conservation at that position, and the height of the letters within each stack indicates the relative frequency of each amino acid at that position.

**Figure 7 genes-13-01455-f007:**
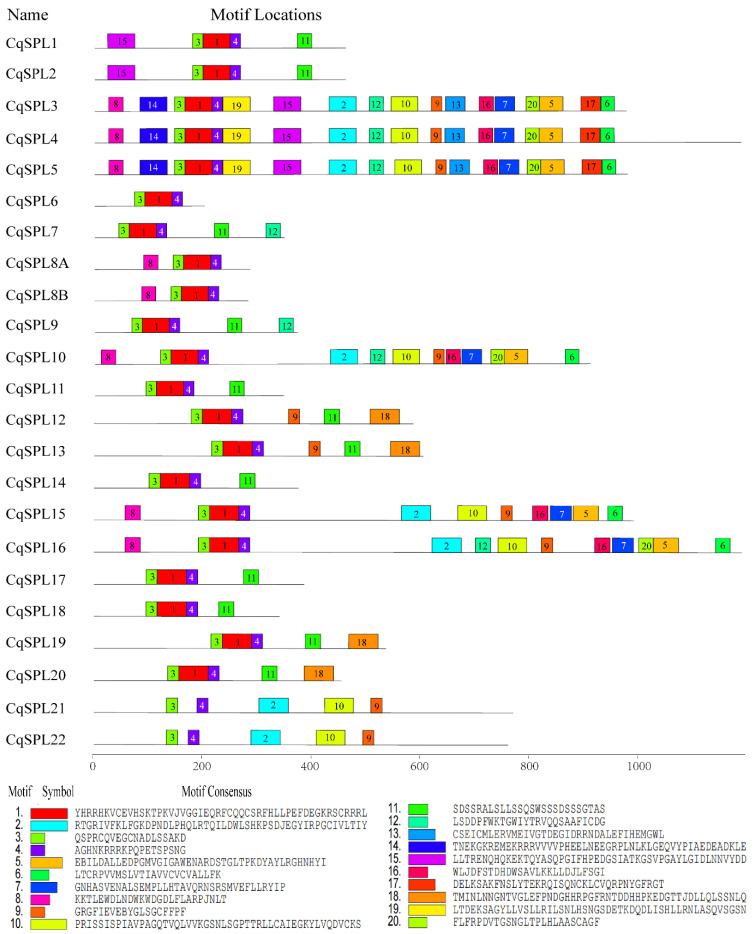
Distribution of conserved motifs in SPL family members of *C. quinoa*. Motifs indicated by different colored boxes were identified using the MEME online tool (meme-suite.org/, accessed on 2 May 2022). A total of 20 motifs were identified, and the conserved SBP-box domain was embedded in the connected motifs 3, 1, and 4.

**Figure 8 genes-13-01455-f008:**
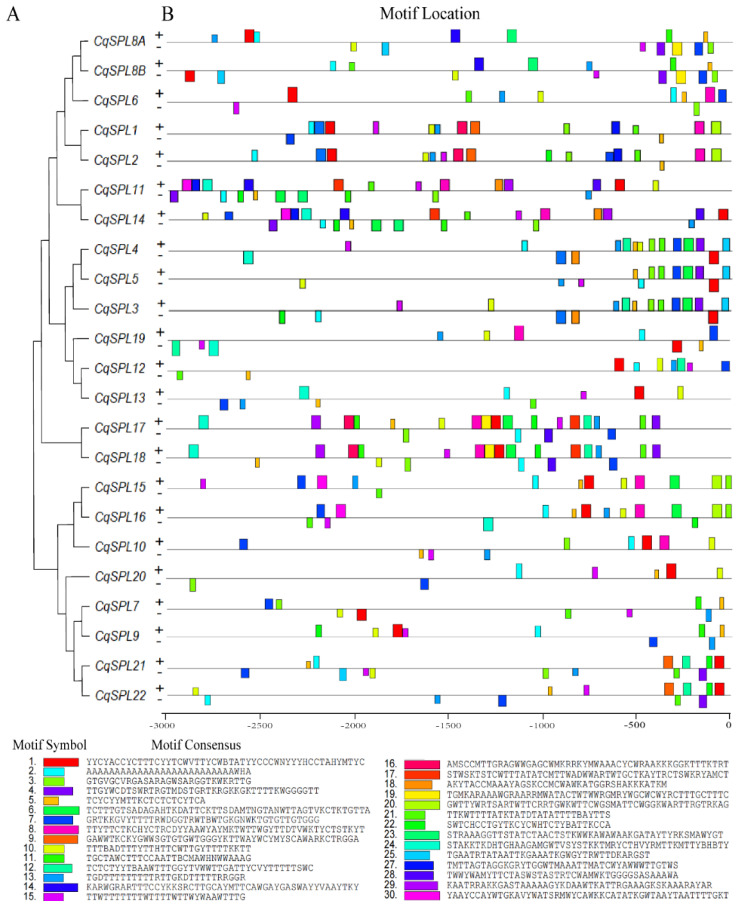
Phylogenetic analysis and motif distribution of *CqSPL* promoters. (**A**) A phylogenetic tree generated based on 3-kb promoters of 23 *CqSPL* genes with MEGA 11 using the neighbor-joining method and 1000 bootstrap replicates. (**B**) The motifs in the promoters were identified using the MEME online tool (meme-suite.org/, accessed on 4 May 2022). The 30 motifs identified are indicated by different colored boxes. The consensus sequence of each motif is shown at the bottom of the figure.

**Figure 9 genes-13-01455-f009:**
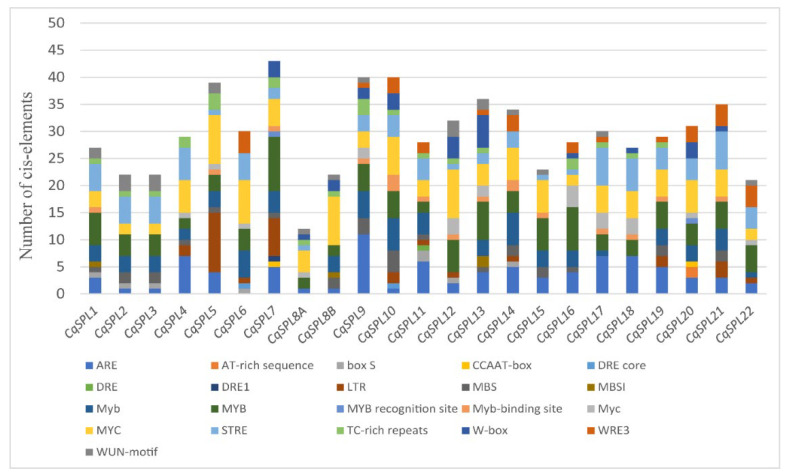
*Cis*-element analysis of *CqSPL* promoters involved in stress responses. *Cis*-elements associated with similar functions are shown in the same color.

**Figure 10 genes-13-01455-f010:**
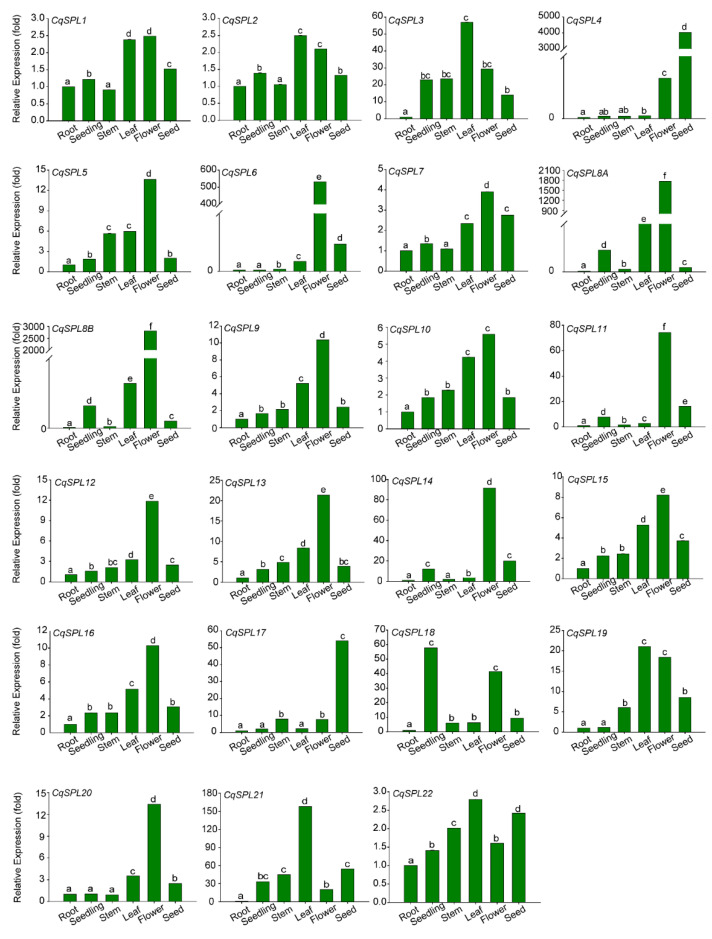
Expression patterns of *CqSPL* genes under normal growth conditions. The expression levels of 23 *CqSPL* genes were determined in different tissues at different developmental stages, and the fold change in the expression level is shown for each *CqSPL*. The expression level in root tissue was arbitrarily set to 1. *CqACTIN2* was used as an internal control. Different letters indicate significant differences in expression.

**Figure 11 genes-13-01455-f011:**
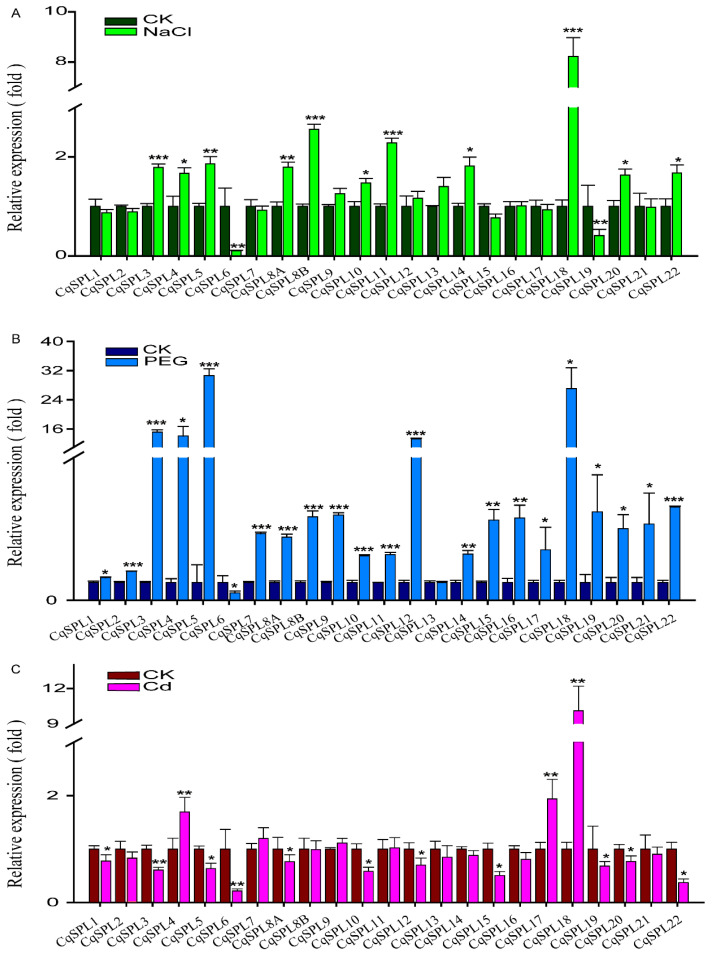
Expression profiles of *CqSPL* genes under salt, drought, and Cd treatments as determined by qRT-PCR. (**A**) 2-h treatment with 200 mM NaCl. (**B**) 2-h treatment with 15% PEG6000. (**C**) 2-h treatment with 100 μM CdCl_2_ (Cd). The relative expression levels of genes were calculated using the 2^−ΔΔCT^ method. The expression level of the control (CK) was arbitrarily set to 1. Error bars indicate standard deviations of the mean value from three biological replicates and three technical replicates. Asterisks indicate significant differences in expression between control and stressed samples (* *p* < 0.05; ** *p* < 0.01; and *** *p* < 0.001).

**Table 1 genes-13-01455-t001:** Molecular features of *SPL* genes identified from the genome of *C. quinoa*.

Gene Name	Gene ID	CDS (bp)	Amino Acids	MW (kD)	pI	Subcellular Localization
*CqSPL1*	AUR62028919	1389	462	50.27	8.90	Nucl
*CqSPL2*	AUR62005629	1389	462	50.31	8.67	Nucl
*CqSPL3*	AUR62002563	2937	978	108.45	6.44	Nucl/Cyto
*CqSPL4*	AUR62029983	3570	1189	132.14	7.01	Nucl/Cyto
*CqSPL5*	AUR62029984	2943	980	108.87	6.30	Cyto/Nucl
*CqSPL6*	AUR62019452	609	202	21.12	9.81	Nucl
*CqSPL7*	AUR62024322	1050	349	37.73	7.64	Nucl
*CqSPL8A*	AUR62004146	849	282	31.95	9.41	Nucl
*CqSPL8B*	AUR62013707	861	286	32.59	9.41	Nucl
*CqSPL9*	AUR62012061	1122	373	39.56	8.45	Nucl
*CqSPL10*	AUR62011728	2955	984	110.16	5.76	Nucl
*CqSPL11*	AUR62029416	1044	347	38.02	8.76	Nucl
*CqSPL12*	AUR62039662	1638	545	60.40	6.26	Nucl
*CqSPL13*	AUR62032118	1689	562	62.08	6.87	Nucl
*CqSPL14*	AUR62003425	1050	349	38.42	8.74	Nucl
*CqSPL15*	AUR62042534	2769	922	101.71	6.75	Nucl
*CqSPL16*	AUR62035190	3321	1106	121.76	7.06	Nucl
*CqSPL17*	AUR62005645	1080	359	39.00	9.36	Nucl
*CqSPL18*	AUR62028905	954	317	34.78	9.05	Nucl
*CqSPL19*	AUR62003075	1500	499	55.39	5.89	Nucl
*CqSPL20*	AUR62007890	1272	423	46.75	6.57	Nucl
*CqSPL21*	AUR62042853	2151	716	80.13	6.32	Nucl
*CqSPL22*	AUR62042654	2124	707	79.25	5.74	Nucl

**Table 2 genes-13-01455-t002:** List of identified miR156/7 loci in *Chenopodium quinoa*.

Name	Chr.	Precursor Length (bp)	Position	Database	Original Name
*Cqu-MIR156a*	4	166	1458734-1458899	PmiREN *	*Cqu-MIR156a*
*Cqu-MIR156b*	4	85	13677430-13677514	PmiREN	*Cqu-MIR156b*
*Cqu-MIR156c*	3	197	75723156-75723352	PmiREN	*Cqu-MIR156c*
*Cqu-MIR156d*	1	86	107403506-107403591	PmiREN	*Cqu-MIR156d*
*Cqu-MIR156e*	2	87	22325967-22326053	This study	
*Cqu-MIR157a*	4	87	6735955-6736041	PmiREN	*Cqu-MIR156e*
*Cqu-MIR157b*	4	82	6736193-6736274	PmiREN	*Cqu-MIR156f*
*Cqu-MIR157c*	4	87	6743900-6743986	PmiREN	*Cqu-MIR156g*
*Cqu-MIR157d*	4	84	6744076-6744159	PmiREN	*Cqu-MIR156h*
*Cqu-MIR157e*	4	87	6786472-6786558	PmiREN	*Cqu-MIR156i*
*Cqu-MIR157f*	4	82	6786710-6786791	PmiREN	*Cqu-MIR156j*
*Cqu-MIR157g*	4	87	6794420-6794506	PmiREN	*Cqu-MIR156k*
*Cqu-IR157h*	4	85	6794596-6794680	PmiREN	*Cqu-MIR156l*
*Cqu-MIR157i*	1	83	115037617-115037699	PmiREN	*Cqu-MIR156m*
*Cqu-MIR157j*	1	87	115046902-115046988	PmiREN	*Cqu-MIR156n*
*Cqu-MIR157k*	1	144	46037964-46038107	PmiREN	*Cqu-MIR156o*
*Cqu-MIR157l*	4	132	52526028-52526159	PmiREN	*Cqu-MIR156p*
*Cqu-MIR157m*	1	85	115031692-115031776	This study	

* Plant MicroRNA Encyclopedia (https://pmiren.com).

## Data Availability

Not applicable.

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
