# Peer review of "Genome-Wide Identification and Characterization of SPL Family Genes in Chenopodium quinoa"

_genes, 2022, doi:10.3390/genes13081455_

Round 1

Reviewer 1 Report

The manuscript entitled; “Genome-wide identification and characterization of SPL family genes in Chenopodium quinoa by Zhao et al. describes about the role of SQUAMOSA PROMOTER BINDING PROTEIN-LIKE proteins in drought and salt tolerance. Experiment is designed well and expression was confirmed using qRT-PCR however, I will suggest following changes.

Abstract

Please explain the objective of current study in the starting lines of abstract

Please provide take-home message in one sentence

Introduction

Please explain hypothesis in the last paragraph of introduction

Materials and Methods

Please provide information about source of light and CO2 levels in the growth chambers

No information was provided about data analysis. Please provide information

Results

Figures quality of all figures must be improved

Expression of CqSPL Genes under Drought, Salt, and Cd Stress

Results must be explained if these are statistically significant or not? For example, expression of gene CqSPL x was increased/decreased significantly under drought stress in comparison to other genes. Please use this style to explain the results of gene expression.

Discussion

Although discussion is quite detailed yet, I can’t see any link among different paragraphs of discussion. Please link each paragraph with connecting sentences/words.

Author Response

The manuscript entitled; “Genome-wide identification and characterization of SPL family genes in Chenopodium quinoa by Zhao et al. describes about the role of SQUAMOSA PROMOTER BINDING PROTEIN-LIKE proteins in drought and salt tolerance. Experiment is designed well and expression was confirmed using qRT-PCR however, I will suggest following changes.

Authors’ answer: Thank you for reviewing our manuscript and your suggestions.

Abstract

Please explain the objective of current study in the starting lines of abstract

Authors’ answer: One sentence “However, there is little information available on SPL genes in Chenopodiaceae” was added in the starting lines of the abstract to point out why we do this study.

Please provide take-home message in one sentence

Authors’ answer: We added “CqSPL genes are thus involved in quinoa responses to salt/drought and cadmium stresses” in the ending lines of the abstract, one sentence message of key finding in this study.  

Introduction

Please explain hypothesis in the last paragraph of introduction

Authors’ answer: We modified the last paragraph of the introduction and tried to explain clearly the goal of this research.

Materials and Methods

Please provide information about source of light and CO2 levels in the growth chambers

Authors’ answer: Added, please see the revised version.

No information was provided about data analysis. Please provide information

Authors’ answer: Added in the section of material and methods, 2.9.

Results

Figures quality of all figures must be improved

Authors’ answer: All figures were saved as TIFs, and the quality of the figures have been improved.

Expression of CqSPL Genes under Drought, Salt, and Cd Stress

Results must be explained if these are statistically significant or not? For example, expression of gene CqSPL x was increased/decreased significantly under drought stress in comparison to other genes. Please use this style to explain the results of gene expression.

Authors’ answer: Thank you for your suggestion. The expression changes of all the genes that were down-regulated or up-regulated under drought, salt, and Cd stress here were significant. We added “significantly” before “responded” in the sentence “Comparison of the expression level between the control and stress-treated seedlings revealed that the expression of 13, 22, and 14 CqSPL genes significantly responded to salt, drought, and Cd treatment, respectively“ to make this clear.

Discussion

Although discussion is quite detailed yet, I can’t see any link among different paragraphs of discussion. Please link each paragraph with connecting sentences/words.

Authors’ answer: Connecting sentences or words were added in the end or the beginning of each paragraph of discussion, please see the revised version.

Reviewer 2 Report

In the current research, the author analyzed the SPL genes family in the crop Chenopodium quinoa, polygenetic analysis, and gene structure analysis, micro-156 targets analysis was performed at the same time, additionally, gene expression profile was also explored.

It is promising that this research provides insights into the Chenopodium quinoa research field, however, some problems should be improved carefully before it could be published in the journal.  The expression of the whole manuscript should be carefully improved, another important thing is that all the figures in the manuscript are not clear, and the figure files should be well organized.

The results part is disordered, for example, it is confusing the author mixes the SPL and micro156 in the first section of the results part, and each part of the results is independent and it is better to combine them together to construct an integrated story here. 

Author Response

In the current research, the author analyzed the SPL genes family in the crop Chenopodium quinoa, polygenetic analysis, and gene structure analysis, micro-156 targets analysis was performed at the same time, additionally, gene expression profile was also explored.

Authors’ answer: Thank you for reviewing our manuscript.

It is promising that this research provides insights into the Chenopodium quinoa research field, however, some problems should be improved carefully before it could be published in the journal.  The expression of the whole manuscript should be carefully improved, another important thing is that all the figures in the manuscript are not clear, and the figure files should be well organized.

Authors’ answer: The whole manuscript was reorganized based on your suggestions and other reviewers’ comments. All figures were resaved as TIFs, and the quality of the figures was improved. We apologize for the disordered figures or tables in the manuscript due to the word file showed different patterns in different computer/laptops.   

The results part is disordered, for example, it is confusing the author mixes the SPL and micro156 in the first section of the results part, and each part of the results is independent and it is better to combine them together to construct an integrated story here.

Authors’ answer: Yes, we agree with you that mixing the CqSPL genes and MIR156/7 in the way of last version might be confusing. The results part thus was rewritten and reorganized with a particular attention to the first section (3.1) of the results part. Please see the revised version.

Reviewer 3 Report

Zhao et al describe in a nice way the identification of the SPL transcription Factor family in the  Chinopodium quinoa genome.

The abstract  summarizes well their work.

Introduction

Between lines 80 and 85 the authors mention why this crop is important but they do not have any references for that. I thnk they should reference those points.

Materials and Methods

In section 2.2 where authors mention web software to perform different analyses, they should mention whether they have used the default parameters or any modifications.

In section 2.4 authors speak about a specific amino acid recognized by the mir156/7 . They should justify that based on reference(s)  or if this is based on their own observations. In the same section, line 149 -150 authors have used the psRNATarget web software. It will be nice to mention if they have used the default parameters. In section 2.5 with the clustalw algorithm, they mention bootstrap but how about the rest of the paraments? The same is also true for sections 2.6 and 2.7, authors should mention whether they have used default parameters

Results

In the results and more specifically in 3.2 section authors say they are trying to identify Ortholog genes, however neither here nor in the materials they mention the methodology or software they have used to do so. There is a plethora ways of doing that an example is here : Sayers EW, Barrett T, Benson DA, Bryant SH, Canese K, Chetvernin V, Church DM, DiCuccio M, Edgar R, Federhen S, et al. Nucleic Acids Res. 2009;37:D5–15.  Basic BLAST is not as robust as specialized software to find Orthologs.

In section 3.4 of the results authors report that they have used three software do check for cis elements . It will be nice to clarify the overlap of their findings and say how many motifs/elements are found from all 3 software . In this way they can report the ones that are more robust and they can do some down stream work probably in the future.

Discussion is well described

Author Response

Zhao et al describe in a nice way the identification of the SPL transcription Factor family in the Chinopodium quinoa genome.

Authors’ answer: Thank you for reviewing our manuscript.

The abstract summarizes well their work.

Authors’ answer: Thank you for your positive feedback.

Introduction

Between lines 80 and 85 the authors mention why this crop is important but they do not have any references for that. I thnk they should reference those points.

Authors’ answer: Two references (61, 62) were added in the revised version.

Materials and Methods

In section 2.2 where authors mention web software to perform different analyses, they should mention whether they have used the default parameters or any modifications.

Authors’ answer: All web softwares mentioned in this section were used with default parameters to perform analysis. This information was added in the section 2.2 of the revised version.

In section 2.4 authors speak about a specific amino acid recognized by the mir156/7. They should justify that based on reference(s) or if this is based on their own observations.

Authors’ answer: Reference (27) was added.

In the same section, line 149 -150 authors have used the psRNATarget web software. It will be nice to mention if they have used the default parameters.

Authors’ answer: We performed the analysis using default parameters, the information was added in the section.

In section 2.5 with the clustalw algorithm, they mention bootstrap but how about the rest of the paraments? The same is also true for sections 2.6 and 2.7, authors should mention whether they have used default parameters

Authors’ answer: We carefully checked the material and methods part, added the parameters or settings for the softwares we used in this study. Please see the revised version.

Results

In the results and more specifically in 3.2 section authors say they are trying to identify Ortholog genes, however neither here nor in the materials they mention the methodology or software they have used to do so. There is a plethora ways of doing that an example is here: Sayers EW, Barrett T, Benson DA, Bryant SH, Canese K, Chetvernin V, Church DM, DiCuccio M, Edgar R, Federhen S, et al. Nucleic Acids Res. 2009;37:D5–15.  Basic BLAST is not as robust as specialized software to find Orthologs.

Authors’ answer: We performed reciprocal best hits using NCBI Blast to find orthologs between SPL genes from both quinoa and Arabidopsis thaliana, this information was added in the materials and methods (2.5).

In section 3.4 of the results authors report that they have used three software do check for cis elements. It will be nice to clarify the overlap of their findings and say how many motifs/elements are found from all 3 software. In this way they can report the ones that are more robust and they can do some downstream work probably in the future.

Authors’ answer: Thank you for your very good suggestion. According to your suggestion, we reexamined the data from the three approaches. Indeed, many motifs/cis-acting elements were revealed by at least two methods, for examples, CCAAT box1, GT1, GATA box, W-box, MYB and MYC related, which might play a role in stress response. But the data presented in last version was mainly from PlantCARE. We rewrote that section, and made it clear in the description of the data. Thank you for your suggestion again, this could help us a lot for our future study.

Discussion is well described

Authors’ answer: We really appreciate for the time you took to review our manuscript, and for your kind suggestions and comments.

Round 2

Reviewer 1 Report

Many thanks for submitting the revised version of the manuscript. Authors have revised the manuscript however; following points are still not tackled.

Abstract

Please explain the objective of current study in the starting lines of abstract

Introduction

Please explain hypothesis in the last paragraph of introduction

Results

Figures quality of all figures must be improved

Expression of CqSPL Genes under Drought, Salt, and Cd Stress

Results must be explained if these are statistically significant or not? For example, expression of gene CqSPL x was increased/decreased significantly under drought stress in comparison to other genes. Please use this style to explain the results of gene expression.

Author Response

Many thanks for submitting the revised version of the manuscript. Authors have revised the manuscript however; following points are still not tackled.

Abstract

Please explain the objective of current study in the starting lines of abstract

Authors’ answer: We appreciated your suggestion, and integrated one sentence into the beginning lines of the abstract in the last version to explain why we do this research/the objective of this study. Please see the responses to reviewers of last version. One sentence “However, there is little information available on SPL genes in Chenopodiaceae” was added in the starting lines of the abstract to point out why we do this study.

Introduction

Please explain hypothesis in the last paragraph of introduction

Authors’ answer: I am sorry, I do not really understand what do you mean here. We carefully rewrote the last paragraph of the introduction according to scientific constraint of the journal and suggestions from other reviewers to make this section logical and clear.  

Results

Figures quality of all figures must be improved

Authors’ answer: All figures were saved as TIFs at 600 ppi, and the quality of the figures has been improved.

Expression of CqSPL Genes under Drought, Salt, and Cd Stress

Results must be explained if these are statistically significant or not? For example, expression of gene CqSPL x was increased/decreased significantly under drought stress in comparison to other genes. Please use this style to explain the results of gene expression.

Authors’ answer: All the genes up- or down-regulated here were with a significant change compared to the control. We explained this to you in the last version of responses to the reviewers. Because we have too many genes to deal with, to avoid repeatedly using “significantly decrease/increase” we still keep the original style in the revised version.  Thank you for your suggestion. The expression changes of all the genes that were down-regulated or up-regulated under drought, salt, and Cd stress here were significant. We added “significantly” before “responded” in the sentence “Comparison of the expression level between the control and stress-treated seedlings revealed that the expression of 13, 22, and 14 CqSPL genes significantly responded to salt, drought, and Cd treatment, respectively“ to make this clear.

Reviewer 2 Report

All the figs should be improved clearly. 

Author Response

All the figs should be improved clearly. 

Authors’ answer: All figures were saved as TIFs at 600 ppi, and the quality of the figures has been improved.